# miR-155-Induced Activation of Pro-Inflammatory Stat1/TBX21 Pathway and M1-Signature Genes Incite Macrophage Apoptosis and Clearance of *Mycobacterium fortuitum* in Zebrafish

Priyanka Mehta [1], Debika Datta [1], Priyanka Dahiya [1] and Shibnath Mazumder [1,2,*]

[1] Immunobiology Laboratory, Department of Zoology, University of Delhi, New Delhi 110007, India
[2] Faculty of Life Sciences and Bio-Technology, South Asian University, New Delhi 110021, India
[*] Correspondence: shibnath@sau.int; Tel.: +91-9999251575

**Abstract:** The role of microRNAs (miRNAs) in *Mycobacterium fortuitum* pathogenesis is not well illustrated. Using zebrafish kidney macrophages (ZFKM) we observed that *M. fortuitum* triggers miR-155 expression, and the TLR-2/NF-κB axis plays a key role in initiating the process. We report that mir-155 activates the pro-inflammatory Stat1/TBX21 pathway in *M. fortuitum*-infected ZFKM. Our results further reveal the role of miR-155 in M1-macrophage polarisation during *M. fortuitum* infection. We observed that miR-155 inhibits *socs1* expression augmenting the expression of *tnf-α*, *il-12* and *ifn-γ* in infected ZFKM. Additionally, attenuating miR-155 inhibited ZFKM apoptosis and increased the intracellular bacterial load implicating its pro-apoptotic and bactericidal role in *M. fortuitum* pathogenesis. This is the first report on the role of miRNA in regulating innate immunity to mycobacteria in fish. We propose that the TLR-2/NF-κB axis triggers miR-155 expression, which in turn represses *socs1* and promotes the development of M1-macrophages. Thus, the functional miR-155/Stat1/TBX21pathway induces a pro-inflammatory milieu favouring ZFKM apoptosis and *M. fortuitum* clearance. Therefore, our study unveils the role of miR-155 in the hierarchy of events leading to *M. fortuitum*-induced apoptosis and bacterial clearance in fish that is still not explored in detail.

**Keywords:** TLR-2; NF-κB; miR-155; *socs1*; M1 macrophages; apoptosis





## 1. Introduction

*M. fortuitum* is an acid-fast, opportunistic non-tuberculous bacterium that infects both cold- and warm-blooded animals [1]. Due to its fast growth and emerging resistance to anti-mycobacterials, *M. fortuitum* is widely accepted as a pathogen of concern [1].

*M. fortuitum* is also a natural fish pathogen and is responsible for piscine mycobacteriosis [2,3] The clinical manifestations of piscine mycobacteriosis are non-specific and include a plethora of symptoms such as ascites, dermal pigmentary changes, loss of scales, granulomas on both external surface and internal organs of the body, and eventually death [2,4]. Although *M. fortuitum* appears to be the major causative agent for piscine mycobacteriosis, other mycobacteria including *M. abscessus*, *M. chelonae*, *M. haemophilum*, *M. marinum* and *M. peregrinum* have frequently been isolated from zebrafish and other fish facilities. The involvement of several unique species often makes the diagnosis and control of piscine mycobacteriosis difficult [4,5]. Recent studies have demonstrated that *M. fortuitum* triggers extensive apoptosis of host macrophages, but the molecular mechanisms remain obscure [6–8].

Pathogen-induced macrophage apoptosis involves the initiation and amplification of both innate and adaptive immune responses [9]. Macrophages provide the first line of defense against mycobacteria by recognizing different conserved mycobacterial ligands with help of pathogen recognition receptors (PRRs), such as the Toll-like receptors (TLRs) [10]. There are also reports suggesting that the macrophages serve as safe havens

for mycobacteria during the dormant period [11]. Among the different TLRs, the role of TLR-2 in regulating mycobacterial-pathogenesis is well known [10]. TLR-2, in conjunction with adaptor proteins myeloid differentiation primary response gene 88 (Myd88) or TIR domain-containing adaptor-inducing interferon-β (TRIF) [12] activates downstream signal transduction pathways leading to activation of pro-inflammatory responses, thereby triggering apoptosis of infected macrophages to contain mycobacterial infection (Mehta et al., 2021). Besides TLR-2, TLR-4 also aids in innate immunity against mycobacterial infections, but the mycobacterial ligand that interacts with TLR-4 remains to be identified [13]. Though the role of TLR-2 in regulating pathogenesis induced by several microbes has been reported in fish [14,15] it has remained less explored in *M. fortuitum* infection.

Among the various post-transcriptional mediators, miRNAs play a critical role in regulating various cellular processes including apoptosis [9,16]. miRNAs are small, single-stranded, evolutionarily conserved, non-coding RNAs that bind to the 3′ UTR of target mRNAs, causing their degradation or inhibition of translation [17]. Compelling evidence suggests the role of miRNAs in reshaping both innate and adaptive immune responses, such that their altered expression is associated with the pathogenesis of diverse diseases, including autoimmunity and cancer [18]. There are reports which have implicated the role of miRNA in controlling microbial infections and pathogens have also been shown to manipulate host miRNA expression to subvert the immune system and establish infections [19]. Although the role of miRNAs in influencing viral immunity is well reported [20], the targeting of host miRNAs by bacterial pathogens is less characterized [21].

Among the various miRNAs that play a role in mycobacterial infections, miR-155 and miR-146 are important [22,23]. It has been observed that miR-155 responds to many inflammatory stimuli, and TLR ligands in various cell types, particularly in monocytes/macrophages. However, studies of the role of miR-155 in regulating macrophage responses to mycobacteria in vitro have generated highly variable and discordant results, probably due to the use of virulent and avirulent mycobacteria strains and to cell lines vis a vis primary monocytic/macrophagic cells [9,22]. The miR-146 family comprises two genes, miR-146a and miR-146b, which are expressed in response to pro-inflammatory stimuli and serve as negative feedback controlling both innate and adaptive immune responses [24]. It has been suggested that miR-146a represses the mycobacteria-induced inflammatory response and facilitates bacterial replication by targeting nitric oxide generation through IRAK/TRAF-6-nitric oxide axis [25]. Nonetheless, it has become increasingly evident that miRNAs play a major role in regulating mycobacterial infections. However, very little is known about the involvement of miRNAs in the pathogenesis induced by *M. fortuitum.*

In the present work, we aimed to identify miRNAs perturbed by *M. fortuitum* in infected macrophages and understand their pathological relevance using adult zebrafish (*Danio rerio*) as the model organism. The cross-talk of innate and adaptive immunity is critical for mycobacterial immunity. Unlike its larval form which has only the innate immune system, adult zebrafish possesses both innate and adaptive immunity resembling that of higher mammals [26] which makes it an attractive model for studying host immunity to mycobacterial infections and extrapolating those to higher vertebrates.

Since miRNAs are essentially conserved across vertebrates, and the zebrafish miRNA family has already been described, we selected it as a convenient model to study miRNA function in mycobacterial disease. We found the expression of several miRNAs to be altered in *M. fortuitum*-infected ZFKM and selected miR-155 for further study. miR-155 is encoded by the *bic* gene (B-cell integration cluster). Specifically, several studies have implicated the role of miR-155 in inducing pro-inflammatory cytokines and orchestrating cell fate decisions in mycobacteria-infected cells [22]. Despite the gamut of literature on miR-155 and mycobacteria cross-talk, there are no studies that suggest its role in piscine-mycobacteriosis. Our results have revealed that miR-155 regulates the expression of pro-inflammatory cytokines thereby positively impacting apoptosis of infected ZFKM and aiding in *M. fortuitum* clearance.

## 2. Materials and Methods

### 2.1. Ethics Statement

Animal experiments described in this study were approved by the Animal Ethics Committee, University of Delhi (DU/ZOOL/IAEC-R/2013/34) and carried out in accordance with the protocols approved by Committee for the purpose of Control and Supervision of Experiments on Animals (CPCSEA), Govt. of India.

### 2.2. Animal Care and Maintenance

Adult zebrafish (*D. rerio*, ZF), 2–3 months old (0.47 ± 0.09 g), were purchased locally (Aquazona, Delhi, India) and maintained in recirculating water tanks at 28 °C under 12 h dark/13 h light conditions. ZF were fed twice a day with commercially available pellets (Taiyo Plus Discovery Special Fish Food) and excess food was always removed from the tanks. The fish were acclimatized for 15 days prior to starting the experiments and during this time water quality parameters (dissolved oxygen content, hardness and pH) and fish health was monitored regularly [27].

### 2.3. Isolation of ZFKM

ZF were sacrificed using an excess of MS 222 (Sigma, New Delhi, India), and kidneys aseptically removed and placed in PBS. Single cell suspensions were prepared by using tissue homogeniser and centrifuging at $400\times g$ for 10 min at 4 °C. The supernatant was discarded, the pellet re-suspended and the RBCs lysed. The phagocyte-rich fraction was allowed to adhere at 30 °C for 2 h, removed by gentle flushing with cold-PBS, washed and further enriched by discontinuous Percoll gradient centrifugation. The purity of ZFKM was checked by staining with Wright-Giemsa stain (85–90%) and viability determined (>95%) using 0.4% trypan blue dye exclusion method. The ZFKM were maintained in RPMI-1640 supplemented with 10% heat-inactivated fetal bovine serum (HiMedia, Mumbai, India) and 1% penicillin-streptomycin (complete-RPMI).

### 2.4. Bacterial Growth Conditions and Infection Assay

*M. fortuitum* (strain 993) was purchased from Microbial Type Culture Collection and Gene Bank (MTCC), Chandigarh, India. Bacteria were grown to mid-log phase (3–4 days) in Middlebrook 7H9 broth (HiMedia, Mumbai, India) at 30 °C supplemented with 0.05% Tween 80, 0.5% glycerol and 100 µg/mL ampicillin with aeration in shaking incubator at 120 r.p.m. The bacteria were harvested by centrifugation, resuspended in RPMI without antibiotics and passed through 26-gauge needle to avoid clumping. ZFKM were infected with *M. fortuitum* at different multiplicity of infection (MOI) to determine the optimal infection dose. A short spin was given to facilitate bacteria-ZFKM interactions. The cultures were incubated at 30 °C for 4 h, washed and treated with amikacin (50 µg/mL) for 1 h. The concentration of amikacin used was able to kill free bacteria but had no effect on ZFKM viability. ZFKM were finally washed, re-suspended in complete-RPMI containing amikacin. For our subsequent studies MOI of 1:10 (ZFKM: *M. fortuitum* i.e., $1 \times 10^7$ CFU) was used for infecting ZFKM.

### 2.5. Inhibitors and miRNA Mimics

ZFKM were pre-treated with *tlr-2* inhibitor (CU-CPT22, 1 µM, Sigma, New Delhi, India), NF-κB activation inhibitor VI (BOT-64, 1 µM, Sigma), and *Casp3a*-specific inhibitor (Ac-DEVD-FMK, 10 µM, Biovision, New Delhi, India) separately for 1 h prior to infection with *M. fortuitum*. The miR-155 mimic (MC11056, Ambion, Life Technologies, New Delhi, India), miR-155 inhibitor (MH11056, Ambion, Life Technologies) and the corresponding controls mirVana miRNA mimic negative control, (4464058, Ambion, Life Technologies), and mirVana miRNA inhibitor negative control, (4464076, Ambion, Life Technologies) were used in this study. Isolated ZFKM were transfected with 30 nM of each oligonucleotide for 16 h using HiPerfect (Qiagen, New Delhi, India). Briefly, 10 µM mimic or inhibitor and 5 µL HiPerfect was added to 90 µL Opti-MEM (Invitrogen, Bangalore, India), incubated for

20 min at 30 °C for complex formation, which was added to ZFKM, maintained in Opti-MEM, and the volume was made up to 1 mL, such that the final concentration of mimic and inhibitor was 30 nM. The ZFKM-mimic or ZFKM-inhibitor complex was incubated at 30 °C under 5% $CO_2$ for 16 h, and subsequently infected with *M. fortuitum.* The doses of the inhibitors, mimics and positive controls were optimized on the basis of their specificity and cytotoxic effects on ZFKM. The indicated doses did not induce cytotoxic effects on ZFKM as determined by trypan blue dye exclusion and were maintained throughout the course of the experiments.

### 2.6. Isolation of Total RNA and Synthesis of cDNA

ZFKM ($1 \times 10^6$ cells/mL) pre-treated with or without specific inhibitors or transfected with miR-155 mimic and miR-155 inhibitor were infected with *M. fortuitum* and the total RNA was isolated at indicated time intervals using TRIzol according to the manufacturer's protocol (Sigma, New Delhi, India). The RNA pellets were washed with 70% ethanol, air-dried and re-suspended in diethyl pyrocarbonate (DEPC) water. RNA concentration and purity were determined (Nanodrop, Thermofisher Scientific, New Delhi, India) and the RNA was further incubated in 1 μL of reaction buffer containing $MgCl_2$ (20 mM) and 1 μL of RNase free DNase I at 37 °C for 30 min to eliminate genomic DNA contamination. The reaction was stopped using 1 μL of EDTA (50 mM) at 65 °C for 10 min. RNA (1 μg) was used as a template for preparing cDNA using the Revert Aid First Strand cDNA Synthesis Kit (Thermo Fisher, New Delhi, India). The cDNA was diluted (1:10) in nuclease-free water and stored at −20 °C for further use.

### 2.7. cDNA Library Preparation

Small RNA libraries for sequencing were constructed at the Genotypic Technology's Genomics facility according to the Illumina TruSeq Small RNA library protocol using 1 μg of total RNA with RIN values 8.4 and 8.7 for control and *M. fortuitum* infected ZFKM, respectively. Briefly, 3′ adaptors were ligated to the specific 3′OH group of micro RNAs followed by 5′ adaptor ligation. The ligated products were reverse transcribed with Superscript III Reverse transcriptase by priming with reverse transcriptase primers. The cDNA was enriched, barcoded by PCR (11 cycles) and cleaned using polyacrylamide gel. The library was size-selected in the range of 140–160 bp, followed by overnight gel elution and salt precipitation using glycogen, sodium acetate (3M) and absolute ethanol. The precipitate was re-suspended in nuclease-free water. The prepared library was quantified using Qubit Fluorometer and validated for quality by running an aliquot on High Sensitivity Bioanalyzer Chip (Agilent, New Delhi, India).

### 2.8. Poly A Tailing and Reverse Transcription

miRNA sequences were extended by poly (A) tailing (New England Biolabs, New Delhi, India), and reverse transcribed into cDNA (Thermo Fisher, New Delhi, India) along with mRNAs in a reverse transcription reaction primed by standard poly (T) adaptor. miR-155-specific primers were designed for amplification of poly (A) tailed miRNAs.

### 2.9. RT–qPCR

ZFKM ($1 \times 10^6$ cells/mL) pre-treated with or without specific inhibitors or transfected with miRNA specific mimic or inhibitors were infected with *M. fortuitum.* ZFKM were harvested at indicated times p.i., total RNA extracted and cDNA prepared. Real-time primers were designed from sequences available in the database and fold changes in gene expression profile were monitored using SYBR green PCR Master Mix (Applied Biosystems (ABI)) by Real-Time PCR (ViiA, ABI). The PCR mixture (total volume 6 μL) contained 3 μL of SYBR AmpliTaq Gold DNA Polymerase (ABI), 1 μL of cDNA, forward and reverse primers (0.2 μL each) and DEPC water (1.6 μL). The list of genes and primers used for RT-qPCR are listed in Table 1. For miR-155-expression analysis RT-qPCR was performed according to the following protocol: pre-activation at 50 °C for 120 s; denaturation at 95 °C

for 10 min; then 40 cycles of amplification at 95 °C for 15 s and 60 °C for 60 s; followed by melting-curve analysis at 95 °C for 5 s, 65 °C for 60 s, and 97 °C for 30 s. For the rest of the genes, RT-qPCR was performed according to the following protocol: denaturation at 95 °C for 20 s; then 40 cycles of amplification at 95 °C for 15 s and 60 °C for 20 s; followed by melting-curve analysis at 95 °C for 15 s, 60 °C for 60 s, and 95 °C for 15 s. The expression of different genes was analysed by the comparative $\Delta\Delta C_T$ method where β-actin was used as the internal control for all the genes and U6 as the endogenous control for miR-155.

**Table 1.** Real-time primer sequences.

| Gene | Sense (5′-3′) | Antisense (5′-3′) | Accession No. |
|---|---|---|---|
| miR-155 | 5′-CGCCGTTAATGCTAATCGTGATAG-3′ | 5′-GCAGGGTCCGAGGTATTCCG-3′ | LM609208.1 |
| *tlr-2* | 5′-ACCTGCTCCAATCTTCAGCTC-3′ | 5′-CTGCTTTCAAGCTCCCGTTC-3′ | NM_212812.1 |
| *tlr-1* | 5′-CGGAGAATCAAGGGAGGTGT-3′ | 5′-TGTGCCGAAGGTTTAGGACT-3′ | NM_001130593.1 |
| *myd88* | 5′-AGTTTGCGCTCAGTCTTTGC-3′ | 5′-ACAGATGGTCAGAAAGCGCA-3′ | NM_212814.2 |
| *irak-4* | 5′-TACTGGACGAGGGTTTTGTGG-3′ | 5′-CGCACTCGAGCTATCCTTCATC-3′ | NM_200163.1 |
| *traf-6* | 5′-ACTAGAGGAGAGCACCCGAG-3′ | 5′-GGAGGACAATAGGCTGACCG-3′ | NM_0010444752.1 |
| *nf-κb* | 5′-AAAAGATGGAGCCCTCACCC-3′ | 5′-ATCAGCCTTGCATCCCTACC-3′ | AY163839.1 |
| *tnf-α* | 5′-TGCTTCACGCTCCATAAGACC-3′ | 5′-CAAGCCACCTGAAGAAAAGG-3′ | NM_212859 |
| *ifn-γ* | 5′-ATGATTGCGCAACACATGAT-3′ | 5′-ATCTTTCAGGATTCGCAGGA-3′ | AB158361 |
| *il-12* | 5′-AGCAGGACTTGTTTGCTGGT-3′ | 5′-TCCACTGCGCTGAAGTTAGA-3′ | AB183001 |
| *il-1β* | 5′-TGGACTTCGCAGCACAAAATG-3′ | 5′-CGTTCACTTCACGCTCTTGGATG-3′ | AY340959 |
| *inos* | 5′-CCAGAGCCTTCGTCTGG GA-3′ | 5′-TTAGAGCCTGGACGAGCGTG-3′ | NM_001104937 |
| *il-6* | 5′-AAGGGGTCAGGATCAGCAC-3′ | 5′-GCTGTAGATTCGCGTTAGACATC-3′ | NM_001261449.1 |
| *cd206* | 5′-TAGTAGGAGCACGACCAGAG-3′ | 5′-GTGAGTGAATGGGACTTGCT-3′ | NM_001310844.1 |
| *arg-1* | 5′-ATCGGCTCAATCTCTGGTCA-3′ | 5′-CAGTCGGTGTGGTTAAAGGT-3′ | NM_001045197.1 |
| *il-10* | 5′-ATTTGTGGAGGGCTTTCCTT-3′ | 5′-AGAGCTGTTGGCAGAATGGT-3′ | NM_001020785 |
| *il-4* | 5′-CATCCAGAGTGTGAATGGGA-3′ | 5′-TTCCAGTCCCGGTATATGCT-3′ | AM403245.2 |
| *stat1* | 5′-TTTTCGTGACTCCTCCACCG-3′ | 5′-AGGATCCGATGCCGCTTTAG-3′ | NM_131480.1 |
| *tbx21* | 5′-ACACTGGCACTCACTGGATG-3′ | 5′-CTCCTTCACCTCCACGATGT-3′ | NM_001170599.1 |
| *socs1* | 5′-AGCACAGAGTTTGAGGTCGC-3′ | 5′-TCTGACAAACTCTCGTCGGC-3′ | AJ972922.1 |
| *Casp3a* | 5′-TTGTCGAGGAACAGAACTGGA-3′ | 5′-GAAGTCTGCTTCAACCGGG-3′ | MG957994.1 |
| *β-actin* | 5′-CGAGCAGGAGATGGGAAC-3′ | 5′-CAACGGAAACGCTCATTGC-3′ | AF057040 |
| *u6* | 5′-TGCTCGCTACGGTGGCACA-3′ | 5′-AAAACAGCAATATGGAGCGC-3′ | NM_001003460.1 |

## 2.10. Enumeration of Intracellular Bacteria

ZFKM ($1 \times 10^6$ cells/mL) transfected with miR-155 mimic and inhibitor were infected with *M. fortuitum*. The ZFKM were harvested at 24 h p.i. for studying the role of miR-155 on regulating intracellular bacterial growth, lysed with 0.1% Triton X-100. MTT [3,-(4, 5-Dimethylthiazol-2-yl)-2, 5 diphenyltetrazolium bromide, 20 μL] (Merck, New Delhi, India) from a stock solution of 5 mg/mL was added and incubated for 5 h. Following incubation, the formazan crystals thus formed were dissolved in dimethyl sulfoxide (DMSO), absorbance read in a microplate reader (Epoch2, BioTek, Bangalore, India) at $A_{595}$. The number of intracellular bacteria was enumerated by interpolating the absorbance of samples from the standard curve.

## 2.11. Apoptotic Studies

### 2.11.1. Annexin V-FITC and Propidium Iodide Staining

The Annexin V-FITC-Propidium iodide (AV-PI) staining was performed using a commercial kit following the manufacturer's instructions (BD-Pharmingen). Briefly, ZFKM ($1 \times 10^6$ cells/mL) pre-treated with specific inhibitors or transfected with miR-155 mimic and miR-155 inhibitor were infected with *M. fortuitum.* The ZFKM were collected at the

indicated time p.i., washed, and stained with AV-FITC -PI mixture and analysed using a Nikon Ti-2 confocal microscope ($\times 100$) within 30 min of adding the dyes. One hundred ZFKM were studied in each field, and three such fields (from 3 experiments) were included to determine the percentage of apoptotic ZFKM. ZFKM were incubated with apoptosis inducer staurosporine (STS, 1 µM, Sigma, New Delhi, India) for 6 h as positive control for the assay.

### 2.11.2. Casp3a Assay

*Casp3a* activity was assayed using the commercial kit following the manufacturer's instruction (Biovision, New Delhi, India). Briefly, ZFKM ($1 \times 10^6$ cells/mL) pre-treated with or without specific inhibitors or transfected with miR-155 mimic and miR-155 inhibitor were infected with *M. fortuitum*. The cells were collected at 24 h p.i., lysed and the supernatant obtained (50 µL) incubated with *Casp3a* specific substrates in reaction buffer for 5 h at 30 °C. The pNA light emission was recorded using a microtiter plate reader at $A_{405}$ nm (Epoch2, BioTek, Bangalore, India) and relative fold changes in *Casp3a* plotted [15].

### 2.12. Statistical Analysis

Mean $\pm$ S.E. was calculated for each parameter used in the present study. The hypothesis was tested using paired Student's *t*-tests assuming unequal variance. To determine the statistical significance, pairwise comparisons were conducted between control (uninfected) and various experimental groups at each time point used in this study. The values for the probability of the null hypothesis (p) less than 0.05 ($p < 0.05$) were considered as statistically significant.

## 3. Results

### 3.1. M. fortuitum-Induced ZFKM Death Is Apoptotic

Our first step was to ascertain the nature of ZFKM death induced by *M. fortuitum*. ZFKM were infected with a range of MOI (ZFKM: bacteria, 1:1, 1:10, 1:25) and viability was checked by trypan blue dye exclusion method at indicated times p.i. for 24 h. Uninfected ZFKM collected at 24 h were used as a control throughout this study. We observed *M. fortuitum* induced ZFKM death to be time- and MOI-dependent (Figure 1A,B) and selected an MOI of 1:10 and 24 h intervals as the standard dose and infection time for further studies.

Mycobacteria-induced cell death could be necrotic or apoptotic. To establish the nature of cytotoxicity induced by *M. fortuitum*, ZFKM were stained with AV-PI and observed under a microscope 24 h p.i. AV staining is indicative of apoptosis while PI is used to identify necrotic cells. In this process of dual staining AV⁺PI⁻ cells are considered early apoptotic, dual positive AV⁺PI⁺ cells are grouped as late apoptotic and AV⁻PI⁺ cells are considered as necrotic cells. Compared to uninfected ZFKM, there were a significant number of AV⁺ ZFKM in the infected cell population (AV⁺PI⁻ 4.67 $\pm$ 0.40%, AV⁺PI⁺ 19.86 $\pm$ 2.58%, AV⁻PI⁺ 2.50 $\pm$ 0.30%) (Figure 1C). STS (positive control) treatment for 6 h showed the presence of a significant number ($p < 0.05$) of apoptotic ZFKM (AV⁺PI⁻ 33.23 $\pm$ 3.20%, AV⁺PI⁺ 10.16 $\pm$ 2.26%, AV⁻PI⁺ 4.03 $\pm$ 1.12%). We observed a maximum number of apoptotic cells (early 4.67 $\pm$ 0.40% and late apoptotic 19.86 $\pm$ 2.58%) and fewer necrotic cells (2.50 $\pm$ 0.30%) at the same time, i.e., 24 h p.i. Hence, we concluded that *M. fortuitum*-induced ZFKM death was primarily apoptotic. These observations align with our previous studies where we reported *M. fortuitum* induces apoptosis of catfish (*Clarias gariepinus* head kidney macrophages) [7,8]. These results established that *M. fortuitum* induces apoptosis of ZFKM.

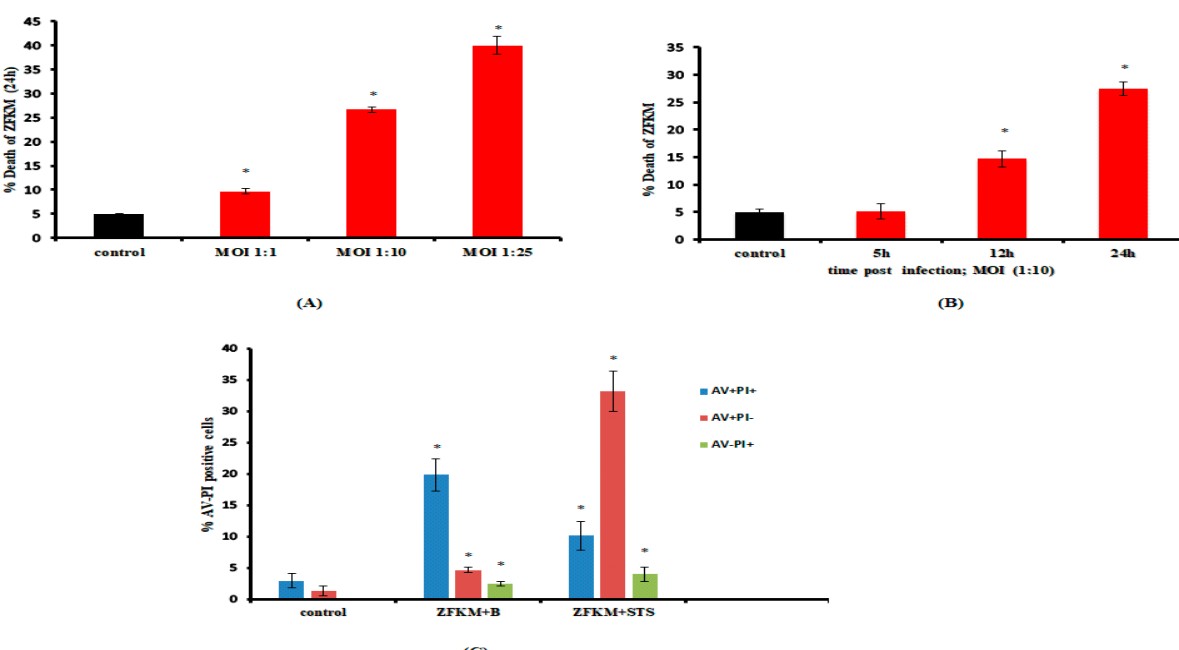

**Figure 1.** *M. fortuitum* induced ZFKM apoptosis is MOI- and time-dependent. (**A**) ZFKM were infected with *M. fortuitum* at different MOI, and percentage death was measured at 24 h p.i. (**B**) ZFKM were infected with *M. fortuitum* (MOI of 10) and cell death was measured at 24 h p.i. (**C**) ZFKM were stained with Annexin V-FITC propidium iodide at 24 h p.i. and apoptosis studied by enumerating AV positive and PI positive ZFKM. Vertical bars represent mean $\pm$ SE ($n = 3$). * $p < 0.05$ compared to control. (ZFKM, uninfected ZFKM; B, *M. fortuitum*).

### 3.2. miR-155 Is Up-Regulated in Response to M. fortuitum Infection

The role of micro-RNAs in *M. fortuitum* pathogenesis is not well studied. At the outset, ZFKM infected with *M. fortuitum* were collected at 5 h p.i. (4 h infection + 1 h amikacin treatment), total RNA was isolated and a miRNA library was created. Hierarchical clustering was accomplished by considering only miRNAs whose profile was significantly modulated in *M. fortuitum*-infected compared to uninfected cells. We observed up-regulation in the expression of 18 miRNAs and down-regulation in 22 miRNAs (Figure 2A). From these up-regulated miRNAs we selected mir-155 for further study.

The next step was validating the RNA-seq data, and for that purpose ZFKM harvested from a different cohort was infected with *M. fortuitum.* The cells were collected at indicated time intervals, cDNA prepared and miR-155 expression monitored. In the absence of avirulent *M. fortuitum* strains, we used formalin-fixed *M. fortuitum* (FF-*M. fortuitum*) as a control for this study. Our RT-qPCR results suggested a time-dependent increase in miR-155 expression with a maximum fold change recorded at 24 h p.i.; (dose—1:10) (Figure 2B) we selected this time point for subsequent studies. Transfection with miR-155 mimic increased the expression of miR-155 while transfection with miR-155 inhibitor repressed the expression of miR-155 in *M. fortuitum*-infected ZFKM (Figure 2C). This was a confirmatory experiment to check the specificity of the miR-155 mimic and inhibitor used for subsequent experiments. Infection with formalin-fixed bacteria (FF-*M. fortuitum*) had no significant effect on miR-155 expression (Figure 2D). These results strongly suggest that miR-155 indeed plays a role in *M. fortuitum* pathogenesis.

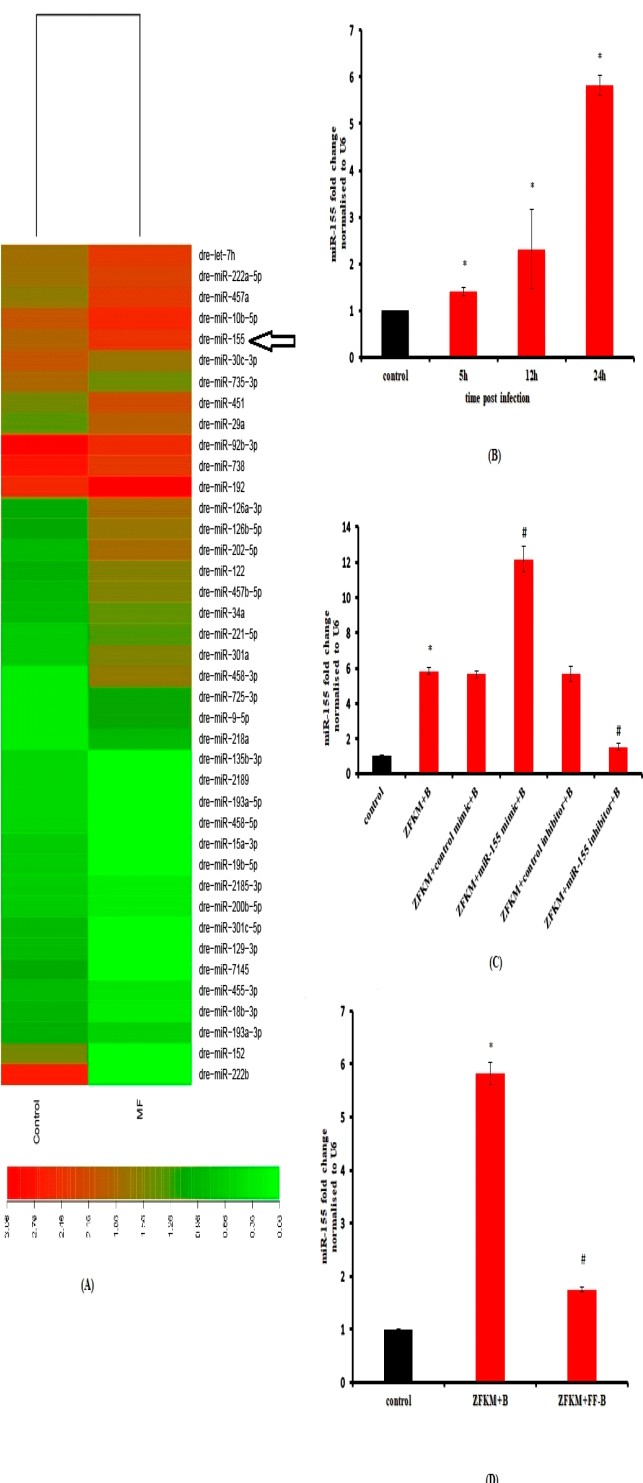

**Figure 2.** miR-155 is up-regulated in *M. fortuitum*-infected ZFKM (**A**) Heat-map depicting altered miRNA expression profile in *M. fortuitum*-infected ZFKM collected 5 h p.i. Arrow indicates miR-155. (**B**) ZFKM were infected with *M. fortuitum* and miR-155 levels quantified at indicated time intervals by RT-qPCR. (**C**) ZFKM transfected with miR-155 mimic or miR-155 inhibitor were infected with *M. fortuitum* and miR-155 levels quantified at 24 h p.i. by RT-qPCR. (**D**) ZFKM were infected with FF-B and miR-155 levels studied by RT-qPCR. The miR-155 data was normalized against U6 snRNA expression in control and infected ZFKM. Vertical bars represent mean ± SE (*n* = 3). * *p* < 0.05 compared to control, [#] *p* < 0.05 compared to ZFKM + B. (ZFKM, uninfected ZFKM; B, *M. fortuitum*; FF-B, Formalin fixed *M. fortuitum*).

### 3.3. tlr-2–irak-4–traf-6–nf-κb Signaling Axis Regulates miR-155 Expression in M. fortuitum-Infected ZFKM

Our next step was identifying the upstream molecules that impact miR-155 expression in *M. fortuitum*-infected cells. We were guided by the documented role of *tlr-2* in mycobacterial pathogenesis [28] and in regulating miR-155 expression [29]. To study this interaction, ZFKM were infected with *M. fortuitum* and the expression of *tlr-2* mRNA was monitored at different time intervals. We observed prolonged expression of *tlr-2* mRNA with a maximum fold change recorded at 5 h p.i. (Figure 3A). TLR-2 agonist Pam3CsK4 (Pam3CsK4, 50 μg/mL, Sigma) was used as a positive control in this study (Figure 3B). TLR-2 acts with TLR-1 as a co-receptor, using Myd88 or TRIF as adaptor molecules. Thus, ZFKM were infected with *M. fortuitum,* and the expression of *tlr-1, myd88* and *trif* was studied at 5 h p.i., as maximum *tlr-2* expression was noted at this time point. We observed significant expression of both *tlr-1* and *myd88* mRNA in *M. fortuitum*-infected ZFKM (Figure 3C). In contrast, no significant change in *trif* expression was observed consequent to *M. fortuitum* infection. Our results confirmed the TRIF-independent coordinated involvement of TLR-2/TLR-1/Myd88 complex in *M. fortuitum* pathogenesis in the infected ZFKM.

We continued to follow the downstream effector molecules in the TLR-2 signaling cascade responsible for inducing miR-155. To this end, we monitored the expression of *irak-4* and *traf-6* in the infected ZFKM and observed a significant fold increase in their expression at 5 h p.i. (Supplementary Figure S1A). To study the link between these molecules, ZFKM were pre-treated with CU-CPT22 then infected with *M. fortuitum,* and the expression of *irak-4* and *traf-6* was quantified by RT-qPCR (Figure 3D). Indeed, significant inhibition of *irak-4* and *traf-6* mRNA expression was observed in the presence of CU-CPT22 which established IRAK-4 and TRAF-6 activation downstream to TLR-2 in *M. fortuitum*-infected ZFKM (Figure 3D).

The TLR-2-IRAK-4-TRAF-6 pathway converges on NF-κB modulating the production of inflammatory cytokines. To this end, we monitored the expression of NF-κB in the infected ZFKM and observed a significant fold increase in their expression at 5 h p.i (Supplementary Figure S1B). Further, ZFKM were pre-treated with CU-CPT22 and then infected with *M. fortuitum* and NF-κB mRNA expression was monitored. The NF-κB activation inhibitor VI (BOT-64) was used as a positive control in this study. We observed a significant increase in *nf-κb* mRNA expression in *M. fortuitum*-infected ZFKM (Figure 3E) that was inhibited in the presence of CU-CPT22 and BOT-64.

The next step was establishing the link between TLR-2 and miR-155 in *M. fortuitum* pathogenesis. To achieve this, ZFKM pre-treated with a TLR-2 specific inhibitor, CU-CPT22, were infected with *M. fortuitum* and the expression of miR-155 was quantified. We observed significant down-regulation of miR-155 expression in the presence of CU-CPT22, suggesting a role of TLR-2 in inducing miR-155 expression in *M. fortuitum*-infected ZFKM (Figure 3F).

We concluded this study by monitoring miR-155 expression in the absence of NF-κB signaling. ZFKM pre-treated with BOT-64 were infected with *M. fortuitum* and the expression of miR-155 was studied by RT-qPCR. We observed that miR-155 expression was significantly down-regulated in the presence of BOT-64 (Figure 3F). Collectively, our results implicate the primal role of the TLR-2-IRAK-4-TRAF-6-NF-κB axis in regulating miR-155 expression in *M. fortuitum*-infected ZFKM.

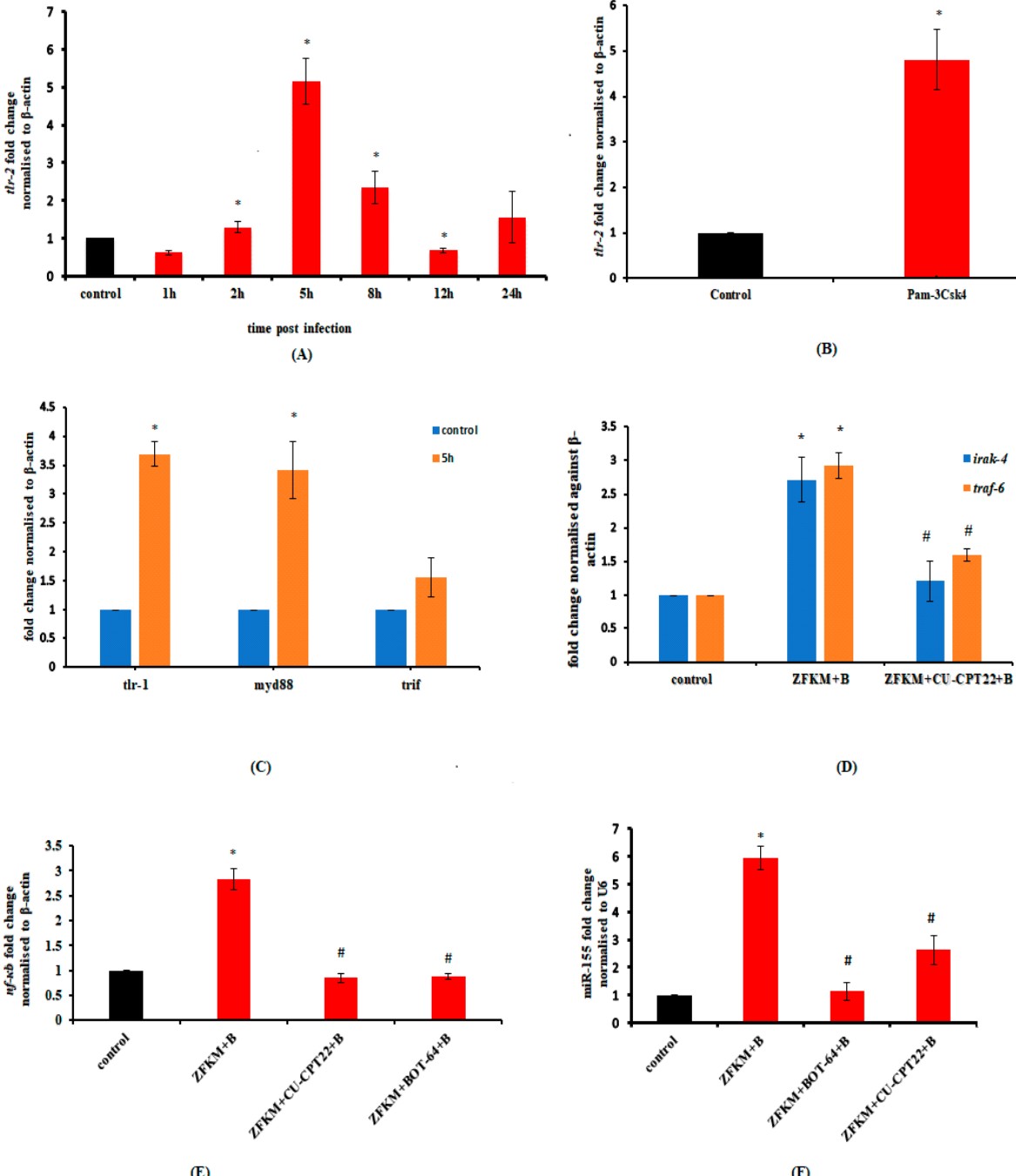

**Figure 3.** The *tlr-2-nf-κb* axis is involved in miR-155 expression. ZFKM were infected with or without (control) *M. fortuitum* and (**A**) changes in *tlr-2* mRNA expression were studied at indicated time intervals, and (**B**) *tlr-2* mRNA expression was studied using *tlr-2* agonist Pam3Csk4 at 5 h p.i by RT-qPCR; (**C**) the changes in *tlr-1*, *myd88* and *trif* mRNA expression were studied at 5 h p.i. by RT-qPCR; (**D**) ZFKM were pre-treated with the indicated inhibitor and infected with *M. fortuitum*, and the changes in *irak-4* and *traf-6* mRNA expression were studied by RT-qPCR. (**E**) ZFKM pre-treated with indicated inhibitor were infected with *M. fortuitum* and the changes in *nf-κb* mRNA expression were studied by RT-qPCR. (**F**) ZFKM pre-treated with indicated inhibitors were infected with *M. fortuitum* and the changes in miR-155 levels were studied at 24 h p.i. by RT-qPCR. Vertical bars represent mean $\pm$ SE ($n = 3$). * $p < 0.05$ compared to control, # $p < 0.05$ compared to ZFKM + B. (ZFKM, uninfected ZFKM; B, *M. fortuitum*; CU-CPT22, *tlr-2* inhibitor; BOT-64, *nf-κb* inhibitor).

*3.4. miR-155 Augments Pro-Inflammatory Cytokine Expression in M. fortuitum-Infected ZFKM*

miRNAs regulate the expression of both pro- and anti-inflammatory cytokines, impacting the host's immunity [30]. To investigate the role of miR-155 in regulating cytokine expression in *M. fortuitum* pathogenesis, we infected ZFKM with *M. fortuitum* and quantified the expression of pro-inflammatory (*ifn-γ* and *tnf-α*) and anti-inflammatory (*il-4*) genes by RT-qPCR. We observed up-regulation in *ifn-γ* and *tnf-α* mRNA with a maximum expression recorded at 12 h p.i. for *ifn-γ* and at 24 h p.i. for *tnf-α* (Figure 4A), and selected these time intervals for further studies. The expression of *il-4* mRNA was found to be comparable to control at all time points in infected ZFKM (Figure 4A). We extended this study by transfecting the ZFKM with a miR-155 mimic and a miR-155 inhibitor before *M. fortuitum* infection and monitored the changes in *ifn-γ*, *tnf-α* and *il-4* mRNA expression. miR-155 overexpression markedly increased *ifn-γ* and *tnf-α* mRNA expression in infected ZFKM (Figure 4B); conversely, transfection with miR-155 inhibitor repressed *ifn-γ* and *tnf-α* mRNA expression. Interestingly, miR-155 played no role in modulating the expression of *il-4* in *M. fortuitum*-infected ZFKM (Figure 4B). Collectively, our results identify the role of miR-155 in promoting pro-inflammatory responses in *M. fortuitum*-infected ZFKM.

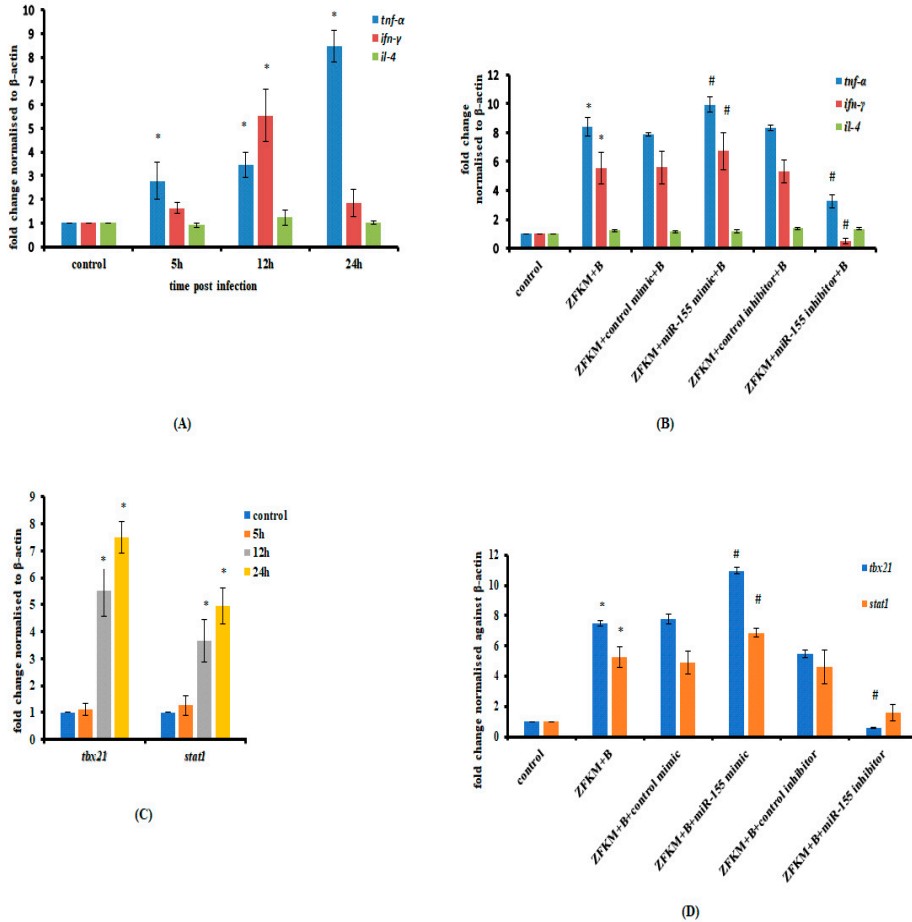

**Figure 4.** miR-155 induces pro-inflammatory responses in *M. fortuitum*-infected ZFKM. ZFKM were infected with *M. fortuitum* and changes in (**A**) *tnf-α*, *ifn-γ* and *il-4* mRNA were studied by RT-qPCR. (**B**) ZFKM transfected with miR-155 mimic and inhibitor were infected with *M. fortuitum* and changes in *ifn-γ* (12 h), *tnf-α* (24 h) and *il-4* (24 h) mRNA expression were studied by RT-qPCR. (**C**) ZFKM were infected with *M. fortuitum* and changes in *tbx-21* and *stat1* mRNA were studied by RT-qPCR. (**D**) ZFKM transfected with miR-155 mimic and inhibitor were infected with *M. fortuitum* and changes in *tbx-21* and *stat1* mRNA were studied at 24 h p.i. Vertical bars represent mean $\pm$ SE ($n = 3$). * $p < 0.05$ compared to control, # $p < 0.05$ compared to ZFKM + B. (ZFKM, uninfected ZFKM; B, *M. fortuitum*).

miRNAs can regulate the expression of transcription factors triggering the development of pro- and anti-inflammatory responses. Therefore, our next step was to correlate miR-155 with the expression of *tbx21* and *stat1* which regulate the development of pro-inflammatory responses [31]. ZFKM were infected with *M. fortuitum* and the expression of these transcription factors was monitored at the indicated time intervals. We observed significant *tbx21* and *stat1*-mRNA expression at 24 h p.i. and selected these time points for further studies (Figure 4C). We then transfected the ZFKM with miR-155 mimic and miR-155 inhibitor and monitored the expression of these transcription factors following *M. fortuitum* infection. Transfection with miR-155 mimic led to a significant increase in *stat1* and *tbx21* mRNA, which was repressed in the presence of miR-155 inhibitor (Figure 4D). Together, these results suggested that a functional miR-155/stat1/tbx21 pathway promotes the development of pro-inflammatory responses in *M. fortuitum*-infected ZFKM.

### 3.5. miR-155 Promotes the Development of M1 Macrophages

Macrophages regulate the development of pro- and anti-inflammatory responses against the invading pathogen. The pro-inflammatory role of miR-155 encouraged us to study its involvement in shaping macrophage-M1/M2 polarization during *M. fortuitum* infection. To begin with, we chose *inos*, *il-1β*, *il-12* and *il-6* as M1 prototype cytokine genes and *cd206*, *arg1*, *tgf-β* and *il-10* as M2 prototype cytokine genes. Time kinetics revealed a maximum up-regulation of *inos*, *il-1β*, *il-12* and *il-6* mRNA at 24 h p.i., and we selected this time point for subsequent studies (Figure 5A). On the other hand, the expression of *cd206*, *arg1*, and *tgf-β* was comparable to the control at all the time points; whereas, the expression of *il-10* was suppressed at all time points in the infected ZFKM (Figure 5A). In the next step, ZFKM transfected with miR-155 mimic and miR-155 inhibitor were infected with *M. fortuitum*, and the expression of M1/M2 signature genes was monitored. It was observed that miR-155 mimic augmented M1 macrophage function by inducing the expression of *il-1β*, *il-12* and *il-6* mRNA and down-regulated the expression of M2-specific *cd206*, *arg1*, *tgf-β*, and *il-10* mRNA expression in *M. fortuitum* infected ZFKM (Figure 5B). The reverse trend was noted following transfection with the miR-155 inhibitor which augmented the expression of *cd206*, *arg1*, *tgf-β* and *il-10* and suppressed the expression of *il-1β*, *il-12* and *il-6* mRNA in *M. fortuitum*-infected ZFKM (Figure 5B). Interestingly, we did not observe any change in the expression of *inos* when treated with either the miR-155 mimic or inhibitor (Figure 5B).

### 3.6. miR-155 Targets socs1 Expression

Among several downstream targets of miR-155, *socs1* is important [32]. We questioned the role of miR-155 in regulating endogenous *socs1* expression in *M. fortuitum*-infected ZFKM. Therefore, the kinetics of *socs1* expression was first studied. We observed a significant fold decrease in *socs1* mRNA expression at 12 h p.i. which gradually increased, reaching the peak levels at 24 h p.i (Figure 6A). In the next step, ZFKM were transfected with miR-155 mimic or miR-155 inhibitor and the expression of *socs1* mRNA was measured. We observed that miR-155 mimic significantly inhibited *socs1* mRNA expression (Figure 6B). Conversely, the expression of *socs1* mRNA was increased in *M. fortuitum* infected ZFKM transfected with miR-155 inhibitor. These results implicated *socs1* to be a target for miR-155 in *M. fortuitum* infected ZFKM.

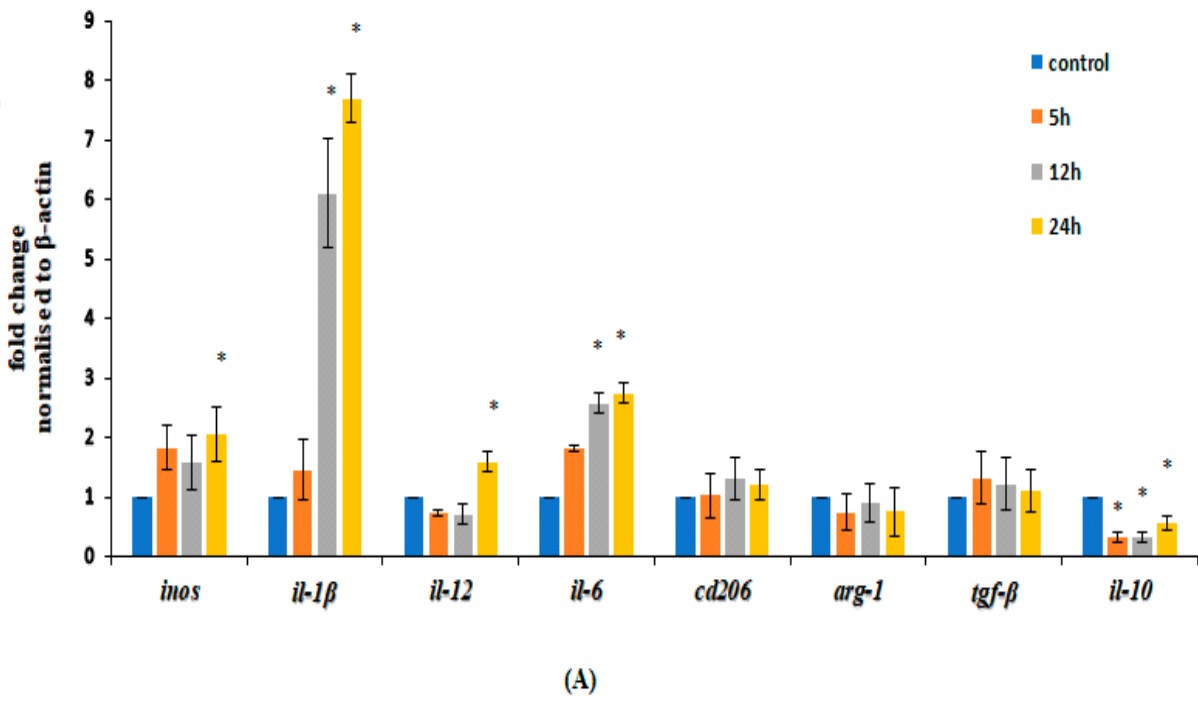

(A)

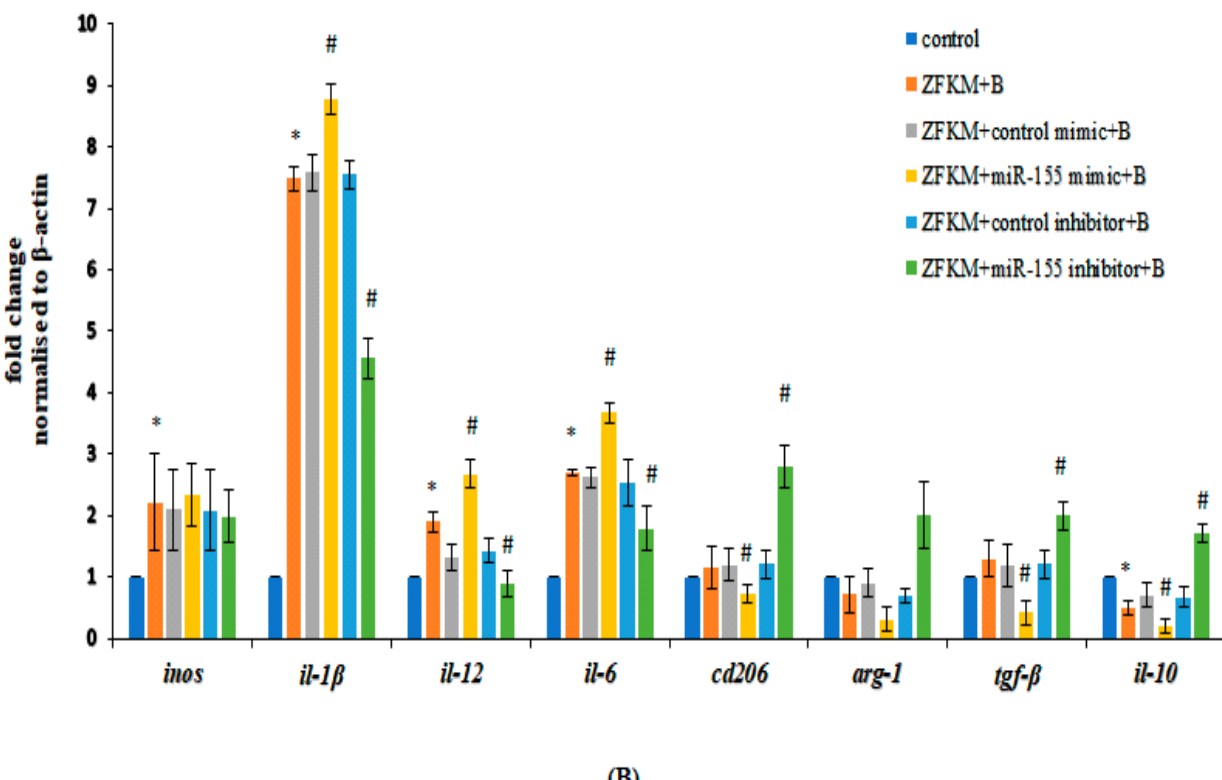

(B)

**Figure 5.** miR-155 favours M1-macrophage polarisation in *M. fortuitum*-infected ZFKM. (**A**) ZFKM were infected with *M. fortuitum* and the expression of M1 (*inos*, *il-1β*, *il-12* and *il-6*) and M2 (*cd206*, *arg-1*, *tgf-β* and *il-10*) signature genes were studied at indicated time intervals by RT-qPCR. (**B**) ZFKM were transfected with miR-155 mimic and miR-155 inhibitor and then infected with *M. fortuitum* and the expression of M1/M2 genes were studied by RT-qPCR at 24 h p.i. Vertical bars represent mean ±SE ($n = 3$). * $p < 0.05$ compared to control, # $p < 0.05$ compared to ZFKM + B. (ZFKM, uninfected ZFKM; B, *M. fortuitum*).

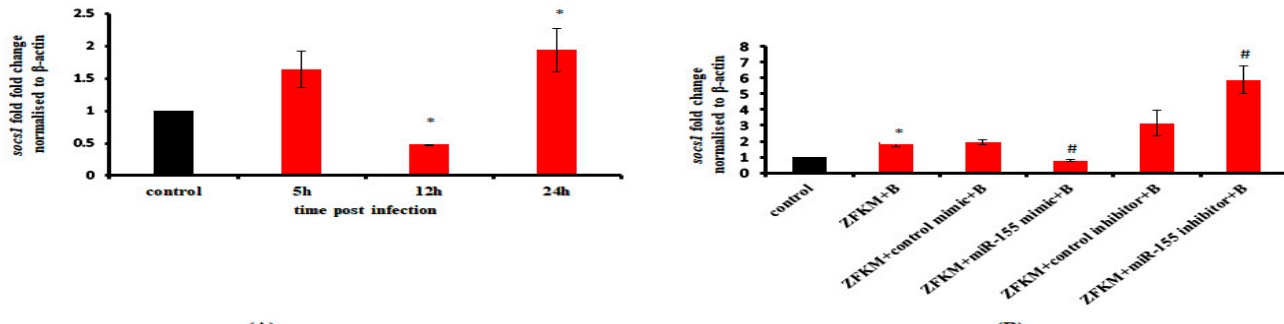

**Figure 6.** miR-155 targets *socs1* expression in *M. fortuitum*-infected ZFKM. (**A**) ZFKM were infected with *M. fortuitum* and *socs1* mRNA expression was studied at indicated time intervals by RT-qPCR. (**B**) ZFKM were transfected with miR-155 mimic or miR-155 inhibitor and then infected with *M. fortuitum* and the expression of *socs1* mRNA was studied at 24 h p.i. by RT-qPCR. Vertical bars represent mean ±SE ($n$ = 3). * $p < 0.05$ compared to control, # $p < 0.05$ compared to ZFKM + B. (ZFKM, uninfected ZFKM; B, *M. fortuitum*).

### 3.7. miR-155 Contributes to ZFKM Apoptosis and Clearance of M. fortuitum

We concluded this study by investigating the role of miR-155 on the growth of intracellular *M. fortuitum*. Accordingly, ZFKM transfected with miR-155 mimic and inhibitor were infected with *M. fortuitum*, and intracellular bacteria were enumerated from cell lysates at 24 h p.i. It was observed that miR-155 overexpression resulted in a significant decline in the number of intracellular *M. fortuitum*, while inhibiting miR-155 expression promoted the growth of intracellular *M. fortuitum* (Figure 7A). Our results suggested that miR-155 signaling restricts the growth of *M. fortuitum* in ZFKM.

Our findings with AV-PI encouraged us to study the role of miR-155 in inducing ZFKM apoptosis and its effect on *M. fortuitum* pathogenesis. Caspase-3 (Casp3a) plays an intrinsic role in the execution of apoptosis. Similarly, we observed a significant increase in *casp3a* mRNA expression in ZFKM consequent to *M. fortuitum* infection (Figure 7B). The next step was exploring the role of miR-155 in *M. fortuitum*-induced ZFKM apoptosis. To study this, ZFKM was transfected with miR-155 mimic and miR-155 inhibitor, and *casp3a* expression and Casp3a activity were studied. We observed that enforced expression of miR-155 resulted in a significant increase in *casp3a* mRNA expression (Figure 7C) and Casp3a protease activity (Figure 7D) in infected cells. Pre-treatment with Casp3a specific inhibitor Ac-DEVD-FMK significantly inhibited Casp3a activity (Figure 7D). Collectively, our results indicate that *M. fortuitum* induces Casp3a-mediated apoptosis of ZFKM.

In concordance with our *Casp3a* results, we noted that overexpression of miR-155 led to a concomitant increase in apoptotic ZFKM (Figure 7E). Further, inhibiting miR-155 expression repressed *Casp3a* activity and a decline in the number of AV+ ZFKM (Figure 7D,F). Thus, we conclude that (1) miR-155 positively regulates *Casp3a* activity triggering apoptosis of *M. fortuitum*-infected ZFKM, and (2) miR-155 induced ZFKM apoptosis is intimately related to the clearance of *M. fortuitum*, thereby impacting the pathogenesis induced by the bacterium.

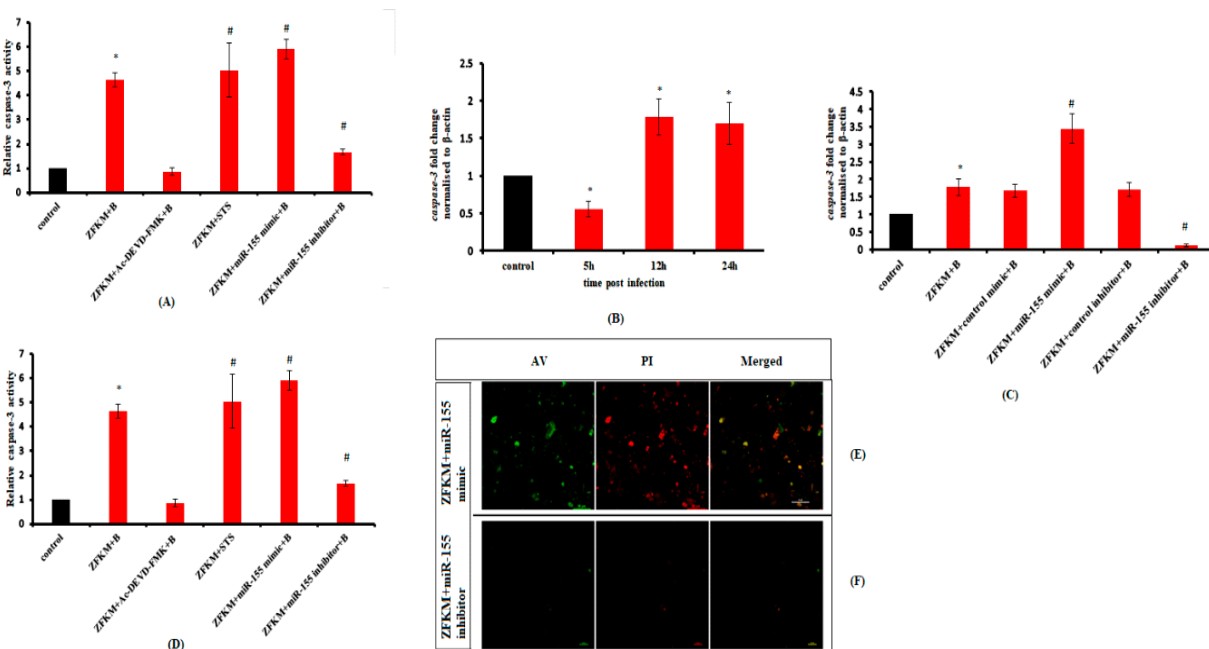

**Figure 7.** miR-155 enhances ZFKM apoptosis leading to *M. fortuitum* clearance. (**A**) ZFKM transfected with miR-155 mimic and inhibitor were infected with *M. fortuitum* and intracellular bacteria enumerated at indicated time intervals. (**B**) ZFKM were infected with *M. fortuitum* and fold change in *Casp3a* mRNA expression was studied at indicated times p.i. by RT-qPCR. (**C**) ZFKM transfected with miR-155 mimic and inhibitor were infected with *M. fortuitum* and *Casp3a* mRNA expression. (**D**) ZFKM transfected with miR-155 mimic and inhibitor were infected with *M. fortuitum* and *Casp3a* activity was studied at 24 h p.i. Ac-DEVD-FMK and STS were used as negative and positive controls, respectively (*Casp3a* activity assay with STS was performed at 6 h p.i.). ZFKM were infected with *M. fortuitum* (**E**) miR-155 mimic and (**F**) miR-155 inhibitor treated ZFKM were stained with Annexin V-FITC propidium iodide at 24 h p.i. and observed under confocal microscope (×100). Scale bar corresponds to 10 μm. * $p < 0.05$ compared to control, # $p < 0.05$ compared to ZFKM + B. (ZFKM, uninfected ZFKM; B, *M. fortuitum:* Ac-DEVD-FMK, *Casp3a* inhibitor; STS, Staurosporine).

## 4. Discussion

The molecular underpinning of *M. fortuitum*-induced macrophage apoptosis is not well understood. Among the different molecular cues that lead to macrophage apoptosis, miRNAs assume significant importance. The present study is the first report demonstrating the major role of miR-155 in *M. fortuitum* pathogenesis in zebrafish.

TLRs play an important role in regulating mycobacterial pathogenesis. To date, how *M. fortuitum*-TLR-2 crosstalk impacts fish immunity is not well understood. Recently, we reported the critical role of TLR-2 in phagocytosis of *M. fortuitum* and in triggering apoptosis of *Clarias gariepinus* macrophages [8]. The genome of *C. gariepinus* has not been sequenced thereby making it difficult to map molecular events that trigger pathogen-induced macrophage apoptosis. To overcome this limitation, we used ZFKM and observed a significant increase in the expression of *tlr-1* and *tlr-2* following infection with *M. fortuitum*. It has been suggested that the TLR-2/TLR-1complex initiates immune signaling cascade using either TRIF or Myd88 as an adaptor [33,34]. We demonstrate *M. fortuitum*-induced TLR-2 signaling to be TRIF-independent; it uses Myd88 as the downstream adaptor for signaling. Further studies are required to identify specific ligands that regulate zebrafish TLR-2/Myd88 complex during *M. fortuitum* infection.

The next step was elucidating the nature of ZFKM death induced by *M. fortuitum* and studying the role of TLR-2 in the process. We observed ZFKM death to be MOI- and time-dependent, implicating intracellular bacterial burden as a major contributor in *M. fortuitum*-induced cell death. Similar findings have been reported previously in *M.*

*tuberculosis* [35], *M. bovis* [36] and *M. smegmatis* [15]. Mycobacteria-induced macrophage death has been reported to be both apoptotic and necrotic in nature [37]. We observed *M. fortuitum*-induced ZFKM death to be apoptotic and caspase-dependent. Additionally, inhibiting TLR-2 attenuated *Casp3a* activation and ZFKM apoptosis, implicating a major role of TLR-2 in *M. fortuitum*-induced macrophage apoptosis. The pro-apoptotic role of TLR-2 has been reported with bacteria representing the Mtb complex, *M. leprae* and several NTM in mammalian systems [38]. The involvement of TLR-2 has also been reported previously in zebrafish with *M. marinum* [39], catfish with *M. smegmatis* [15] and *M. fortuitum* [8] together, this implicates TLR-2 to be an evolutionarily conserved mycobacterial PRR. Besides intracellular bacterial load, other factors also play role in mycobacteria-induced apoptosis [40]. Although we are not sure about the role of different molecular intermediates in *M. fortuitum*-induced ZFKM apoptosis, the crosstalk of TLR-2-dependent cytokines like TNF-$\alpha$ could be important [41].

The role of miRNA in mycobacterial-pathogenesis is well accepted with reports suggesting TLR-2-miRNA crosstalk plays a major regulatory role in the process. Zebrafish-miRNAs have been used successfully to map various diseases, but there are no reports on *M. fortuitum*-induced pathogenesis. In the absence of prior information, we initially created a miRNA library in ZFKM upon infection with *M. fortuitum*. Our transcriptome data suggested significant changes in the expression of several miRNAs, and we selected miR-155 for the present study. miR-155 is known as a prototype multifunctional miRNA that is induced in macrophages and dendritic cells [42]. It serves as an initial immune sensor, modulating both innate and adaptive immune responses to a wide range of infections including mycobacteria [9]. Our RT-qPCR studies confirmed a time-dependent increase in miR-155 expression in *M. fortuitum*-infected ZFKM. These results are consistent with previous reports suggesting time-dependent expression of miR-155 expression in mycobacteria-infected mouse macrophages [9].

The role of TLR-2 in miR-155 expression being well reported in mycobacterial infection [9] made us excited to link the two molecules in *M. fortuitum* infection. We report TLR-2-dependent up-regulation of miR-155 via Myd88 in ZFKM during *M. fortuitum* infection. Incidentally, dead *M. fortuitum* failed to trigger miR-155 expression. Live and dead mycobacteria each trigger distinct signaling pathways with different outcomes on the host macrophages [43]. Our results clearly suggest live *M. fortuitum* to be the stimulus for inducing miR-155 expression and downstream signaling cascade. Identifying the signaling cascades triggered by live and dead *M. fortuitum* is important for understanding the pathogenesis induced by the bacterium.

TLR-2/Myd88-dependent signaling requires downstream involvement of several intermediates. We selected IRAK-4 and TRAF-6 for this study. IRAK-4, an essential component of the TLR-2/Myd88-dependent pathway, functions both as kinase and adapter activating subsets of divergent signaling pathways. Its association with TRAF-6 leads to activation of the classical NF-$\kappa$B-dependent inflammatory genes. Our results suggested TLR-2-dependent activation of the IRAK-4-TRAF-*6* axis in *M. fortuitum*-infected ZFKM. Previous studies have suggested the role of NF-$\kappa$B in miR-155 expression and *vice versa* [44]. We observed that pharmacological inhibition of TLR-2 and NF-$\kappa$B repressed miR-155 in infected ZFKM; suggesting increased expression of miR-155 to be ascribed to the activation of the TLR-2 and NF-$\kappa$B signaling pathway by live *M. fortuitum*. Collectively, our results firmly established that TLR-2/Myd88-IRAK-4-TRAF-6 converges at NF-$\kappa$B to induce miR155 expression in ZFKM during *M. fortuitum* infection.

Both NF-$\kappa$B and miR-155 regulate a large number of pro-inflammatory genes; hence, their crosstalk is a major event in maintaining immune homeostasis, though the mechanisms remain unclear. A key signaling hub that integrates downstream TLR-2/Myd88-IRAK-4-TRAF-6 signaling pathways for NF-$\kappa$B activation is the formation of transforming growth factor-$\beta$-activated kinase 1 (Tak1) and TAK1 binding protein 1 (Tab1) complex. Following activation, the Tak1/Tab1 complex activates downstream kinase IKK, which in turn phosphorylates the NF-$\kappa$B inhibitor I$\kappa$b$\alpha$ leading to ubiquitin-dependent I$\kappa$b$\alpha$

degradation and NF-κB activation. It has been observed that miR-155 plays an important role in the activation of Tak1/Tab1 signalosome leading to translocation of NF-κB into the nucleus and transcription of NF-κB-dependent genes [45]. On the other hand, there are also reports suggesting NF-κB activates the transcription of miR-155 [44]. Our results support previous studies suggesting transcription of miR-155 is regulated by NF-κB. Although it is not possible from this study to conclude how NF-κB regulates the miR-155 expression, we suggest that NF-κB binds to the promoter region of the miR-155 gene to activate its transcription and trigger a chain of events. Unraveling the NF-κB-mediated regulation of miR-155 will help in understanding *M. fortuitum* pathogenesis and designing targeted therapies. Although the role of TLR-2/Myd88-NF-κB axis in enhancing miR-155 expression has been reported earlier in mammalian macrophages [46] to the best of our knowledge this is the first report in fish.

miRNA modulates the development of opposing pro- and anti-inflammatory responses, thereby impacting host immunity [47]. We observed that overexpression of miR-155 augmented the expression of pro-inflammatory *ifn-γ* and *tnf-α* and repressed anti-inflammatory *il-10* expression while the reverse trend was noted on inhibiting miR-155 expression. Incidentally, we did not observe any effect of miR-155 on key anti-inflammatory *il-4* expression in *M. fortuitum*-infected ZFKM, and this requires further investigations. Our study suggests miR-155 tilts the balance in favour of pro-inflammatory cytokine expression in *M. fortuitum*-infected ZFKM. The exact evidence of Th1/Th2 differentiation has not been completely established in fish [48], but the up-regulation of *ifn-γ* and *tnf-α* expression subsequent to the *M. fortuitum* infection and the correlation with miR-155 both indicate the role of miR-155 with Th1 regulation in fish.

Further, our results also demonstrated that miR-155 overexpression led to a concomitant decrease in intracellular bacterial growth implying an inverse correlation between miR-155 and *M. fortuitum* growth. We had previously reported the role of pro-inflammatory cytokines in inducing HKM apoptosis and removal of *M. smegmatis* [15] and *M. fortuitum* [8] by fish macrophages. Based on our current findings we extend our previous results and propose that miR-155-mediated production of pro-inflammatory cytokines helps in the removal of infected ZFKM thereby preventing *M. fortuitum* dissemination. Our results are also in accordance with previous studies suggesting macrophages infected with mycobacteria undergo apoptosis preventing its diffusion [16].

The development of M1 and M2 macrophages is contentious. The M1 macrophages are primarily involved in provoking pro-inflammatory responses while the M2 macrophages have anti-inflammatory and reparative functions. Recent studies have also suggested the role of miRNAs in macrophage polarization [49]. Based on the inflammatory spectrum and enhanced miR-155 expression observed in infected ZFKM, we hypothesized the role of miR-155 in M1 polarisation during the macrophage inflammatory state. To test this, we studied the expression of *inos*, *il-1β*, *il-12* and *il-6* for M1 and the expression of *cd206*, *arg1*, *tgf-β* and *il-10* for studying M2 responses. Our results demonstrated a transition of ZFKM towards the M1 phenotype during *M. fortuitum* infection with miR-155 playing a positive role in the process. Incidentally, though significant *inos* expression was recorded in *M. fortuitum*-infected ZFKM our mimic and inhibitor studies suggested little role of miR-155 on *inos* expression. These results are in contradiction to previous studies suggesting the role of miR-155 on *inos* expression [14,50]. Response to a pathogen is often host-specific and at this stage we are not sure whether this represents a fish-specific innate response to mycobacteria and needs further studies. The precise molecular events that lead to nitric oxide production in *M. fortuitum*-infected ZFKM are not known. Among different upstream molecules that influence the NO pathway, nucleotide-binding oligomerization domain-containing protein 2 (NOD2) is important. NOD2 is a cytosolic PRR that regulates macrophage-inflammatory responses under a variety of stress including microbial infections [51]. Previous studies have linked NOD2 with NO production in mycobacteria-infected macrophages entailing NF-κB signaling [52]. Future studies aimed to understand the role of NOD2 in *M. fortuitum*-

induced NO production will throw more light towards understanding the pathogenesis induced by the bacterium.

The next step was exploring the miR-155-induced M1 switch. We argued that miR-155 mediates its effects via transcriptional regulation. Indeed, we observed that miR-155 enhanced the expression of *stat*1 and *tbx21*, the key pro-inflammatory transcription factors. It has been observed that miR-155 regulates the expression of *stat1*-dependent *tbx21* via several intermediaries including suppression of cytokine signaling 1 (*socs1*) [30,53]. SOCS1 functions as a molecular switch tuning JAK-STAT signaling pathways and activates cytokine-receptor complexes to regulate the functioning of various immune cells including macrophage-inflammatory processes [32]. We report that inhibiting miR-155 augmented *socs1* expression, suggesting *socs1* to be a downstream target of miR-155 in *M. fortuitum*-infected ZFKM. *socs1* also functions as a pro-survival molecule inhibiting apoptosis of various cell types [32]. Our results implicated that inhibiting miR-155 signaling not only augmented *socs1* expression it also suppressed pro-inflammatory cytokine expression and inhibited apoptosis of the infected ZFKM. Although our results are in line with previous studies suggesting the role of *socs1* in mir-155-mediated immune regulation [54], more confirmatory studies are needed to validate that mir-155-mediated repression of *socs1* impacts *M. fortuitum* pathogenesis in zebrafish.

## 5. Conclusions

Our study provides insights into miRNA-target interactions in regulating *M. fortuitum* pathogenesis in fish. We report that TLR-2 plays a critical role in integrating bacterial stimuli with apoptotic signals in *M. fortuitum*-infected macrophages. TLR-2 signaling converges at NF-κB-mediated induction of miR-155, suppressing *socs1* to augment pro-inflammatory cytokine production, which leads to apoptosis of infected macrophages and pathogen clearance. *M. fortuitum* exhibits diverse host tropism, and we strongly believe that studying zebrafish-*M. fortuitum* interactive pathways will shed important light on the universal aspects of these regulations, beyond piscine mycobacteriosis which may not be possible in other systems such as mammals.

**Supplementary Materials:** The following supporting information can be downloaded at: https://www.mdpi.com/article/10.3390/microbiolres14020039/s1.

**Author Contributions:** Conceived and designed the experiments: P.M., D.D., P.D. and S.M; performed the experiments: P.M., D.D. and P.D.; analyzed the data: P.M., P.D. and S.M.; contributed reagents/materials/analysis tools. All authors have read and agreed to the published version of the manuscript.

**Funding:** This work was supported in part by the RRG fund from South Asian University (Grant Nos: RRG/2019-20) and by the Indian Council for Agricultural Research (Grant Nos: NFBSFARA/ICAR RNAi-2014). The funders had no role in the study design, data collection and analysis, decision to publish, or preparation of the manuscript. P.M. and P.D. were supported by the UGC Fellowship (Govt. of India). D.D. was supported by the ICAR Research Fellowship (Govt. of India).

**Institutional Review Board Statement:** Animal experiments described in this study were approved by the Animal Ethics Committee, University of Delhi (DU/ZOOL/IAEC-R/2013/34) and carried out in ac-cordance with the protocols approved by Committee for the purpose of Control and Supervision of Experiments on Animals (CPCSEA), Govt. of India.

**Data Availability Statement:** Required details have been added in the manuscript.

**Acknowledgments:** We thank Beena Pillai and her student Divya Chaubey, IGIB-CSIR, New Delhi for the help in providing reagents and in performing this study, and Ayub Quadri, NII, New Delhi for providing Pam3Csk4. We are grateful to Daman Saluja, ACBR, University of Delhi and Milind Dongerdive, SAU, New Delhi for the help in performing the RT-qPCR studies. The authors acknowledge the help from Aiyas and his team at Genotypic Technologies for help in the miRNA library work.

**Conflicts of Interest:** The authors declare that no competing interest exists.

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
