# Peer review of "miR-155-Induced Activation of Pro-Inflammatory Stat1/TBX21 Pathway and M1-Signature Genes Incite Macrophage Apoptosis and Clearance of Mycobacterium fortuitum in Zebrafish"

_2036-7481, doi:10.3390/microbiolres14020039_

Round 1

Reviewer 1 Report

The authors examine a very interesting topic the role of microRNA in the pathogenesis of Mycobacterium fortuitum. They found that miR-155 control the expression of the proinflammatory cytokines thus contributing to apoptosis of the infected zebrafish macrophages and clearance of M. fortuitum. In particular they found that TLR-2 play a crucial role in the regulation of miR-155 through Myd88 during M. fortuitum infection.

The publication is neatly structured, well planned and includes carefully carried out experiments. The results are convincing and very well presented.  The introduction and the discussion are wide and well articulated.  Therefore, I  strongly recommend this manuscript for publication in Microbiology Research

Author Response

We thank the reviewer for finding the work very interesting and strongly recommending the manuscript for publication.

Reviewer 2 Report

In this manuscript the role of microRNAs (miRNAs) in Mycobacterium fortuitum-pathogenesis is analyzed by using zebrafish kidney macrophages (ZFKM). They convincingly show that M. fortuitum triggers miR-155 expression and the TLR-2/NF-κB axis plays a key role in initiating the process, and that mir-155 activates the pro-inflammatory Stat1/TBX21 pathway in M. fortuitum-infected ZFKM. 

Moreover, they identified a role of miR-155 in M1-macrophage polarization during M. fortuitum infection and observed that miR-155 inhibits socs1 expression augmenting the expression of tnf-α, il-12 and ifn-γ in infected ZFKM.

Additionally, they show that attenuating miR-155 inhibited ZFKM apoptosis and increased the intracellular bacterial load, implicating its pro-apoptotic and bactericidal role in M. fortuitum-pathogenesis.

Conclusions are sound and methods are adequately described and performed. Results are interesting and provide some insights in the hierarchy of events leading to M. fortuitum-induced apoptosis and bacterial clearance in fish.

Minor concerns

Figures are poor quality, and they must be improved.

Panel A and B of figure 3 are not labeled.

Author Response

We thank the reviewer for the critical comments, which helped improve the manuscript's overall quality. In the revised manuscript, we have improved the figures' quality and properly labeled Fig. 3. 

We hope to have addressed your concerns and that you will accept our manuscript. 

Reviewer 3 Report

The authors study the impact of miR-155 in protection against infection of M. fortuitum in ZFKM cells. While the authors assess different factors, conclusions to some results within the study are misleading. Further, some controls are missing for some experiments and the figures submitted herein are difficult to assess due to their low resolution and pixelation. Below are comments highlighting some of these points.

Comments

Figure 1C suggests that M. fortuitum induces both necrotic and apoptotic cell death, given that the largest number of ZFKM belonged to the AV+PI+ category. Further, there is approximately a 2-fold change between AV+PI- and AV-PI+ ZFKM infected with mycobacteria, which is not a significant difference to conclude that M. fortuitum only induces apoptotic cell death. As such, the conclusion in the results section for this figure is misleading and should be adjusted.

As mentioned previously, the quality of the figures is poor making it difficult to assess and understand the data.

What is the control mimic, miR155-mimic, control inhibitor, and miR155 inhibitor in Figure 2C? This experiment is also missing a control that measures RNA levels of a gene of M. fortuitum as a proof of infection.

All the controls in all the panels of Figure 3 need to be clearly labelled in each figure. Further, what are the timepoints for each of the inhibitors plotted in Figures 3D and 3E? Conclusions to these panels are misleading given that they lack the same timepoints collected for both treated and untreated conditions.

Why are there two different timepoints on the same figure of Figure 4B? All the factors monitored would be best represented at the same timepoints. Further the inhibitor does not significantly increase gata-3 and klf-4 levels compared to the control (the increase was ~2-fold in Figure 4D). As such, concluding an effect of the inhibitor on these two factors is misleading when looking at the data in Figures 4C and 4D.

Conclusions for Figures 7D and 7E are also misleading, given that the merged confocal microscopy image appears to have almost overlapping AV-stained and PI-stained cells. As mentioned earlier, this further suggests that M. fortuitim induces both necrotic and apoptotic cell death rather than selectively apoptosis. Similar conclusions were inferred in the discussion, which are not supported by the data in Figures 1 and 7.

Author Response

We sincerely thank you for spending valuable time critically reviewing the manuscript. Your comments were very helpful in improving the quality of our work. We have answered all your questions and hope the explanations will suit you, and that the manuscript is accepted.

1. Figure 1C suggests that M. fortuitum induces both necrotic and apoptotic cell death, given that the largest number of ZFKM belonged to the AV+PI+ category. Further, there is approximately a 2-fold change between AV+PI- and AV-PI+ ZFKM infected with mycobacteria, which is not a significant difference to conclude that M. fortuitum only induces apoptotic cell death. As such, the conclusion in the results section for this figure is misleading and should be adjusted.
We appreciate the concern raised by the reviewer. In the present study we have used annexin V and propidium iodide (AV-PI) staining to determine the nature of cell death induced by M. fortuitum at 24 h p.i. AV staining is indicative of apoptosis while PI is used to identify necrotic cells. In this process of dual staining AV+PI - cells are considered early apoptotic, dual positive AV+PI+ cells are grouped as late apoptotic and AV-PI+ cells are considered necrotic cells. We observed the maximum number of apoptotic cells (early and late apoptotic, 24.53 % ) and few necrotic cells 2.50 %  at the same time point i.e. 24 h p.i. and concluded that M. fortuitum-induced ZKM death to be primarily apoptotic in nature. These observations are in line with our previous studies, where we reported M. fortuitum induces apoptosis of catfish (Clarias gariepinus head kidney macrophages) (Datta, D., Khatri, P., Singh, A., Saha, D. R., Verma, G., Raman, R., & Mazumder, S. (2018). Mycobacterium fortuitum-induced ER-Mitochondrial calcium dynamics promotes calpain/caspase-12/caspase-9 mediated apoptosis in fish macrophages. Cell death discovery, 4, 30. https://doi.org/10.1038/s41420-018-0034-9); (Dahiya, P., Hussain, M. A., & Mazumder, S. (2021). mtROS Induced via TLR-2-SOCE Signaling Plays Proapoptotic and Bactericidal Role in Mycobacterium fortuitum-Infected Head Kidney Macrophages of Clarias gariepinus. Frontiers in immunology, 12, 748758. https://doi.org/10.3389/fimmu.2021.748758)

2. As mentioned previously, the quality of the figures is poor making it difficult to assess and understand the data.
We are thankful to the reviewer for pointing this out. In the revised manuscript, we have improved the quality of the figures, and we hope they address your concern and that the paper will be accepted.
 3. What is the control mimic, miR155-mimic, control inhibitor, and miR155 inhibitor in Figure 2C? This experiment is also missing a control that measures RNA levels of a gene of M. fortuitum as a proof of infection.

Control mimic and control inhibitor are validated random sequences tested in the living systems and are shown to produce no identifiable effects on miRNA function. 
A control mimic and inhibitor study was performed for the correct interpretation of results and to confirm that the experimental system is working as expected. In our study, control mimic and inhibitor are used as negative control and are shown to produce no identifiable effects on known miRNA function. 
A negative control is used to indicate if the results are nonspecific. Comparison of results from the negative control with results from the miRNA-specific mimic or inhibitor under study can be used to confirm that the observed results are specific to the miRNA mimic or inhibitor under investigation.
miR-155 mimics are small, chemically modified double-stranded RNA molecules designed to specifically bind to and mimic endogenous miR-155 and enable miRNA functional analysis by up-regulation of miRNA activity. On the other hand, miRNA 155 inhibitors are natural or artificial RNA transcripts that sequestrate miR-155 and decrease or even eliminate their effects. These are primarily used to study function by targeting mRNA as an exogenous tool.

We are thankful for your suggestion of measuring RNA levels of a gene as proof of infection. This is a good suggestion, but we are currently not sufficiently trained for this study. Please note that in lieu of performing such real-time assays to measure bacterial infection, we have measured intracellular bacterial replication as proof of M. fortuitum infection. We would train and perform similar studies in our future work.  

4. All the controls in all the panels of Figure 3 need to be clearly labelled in each figure. Further, what are the timepoints for each of the inhibitors plotted in Figures 3D and 3E? Conclusions to these panels are misleading given that they lack the same timepoints collected for both treated and untreated conditions.

 We thank the reviewer for pointing this out. In the revised manuscript, we have properly labeled all the controls.  
Please note that in Fig. 3D, we initially did a time kinetics study and observed the maximum expression of irak-4 and traf-6 at 5 h p.i. and selected this time point for inhibitor studies. Thus, ZFKM were treated with the indicated inhibitor, and irak-4 and traf-6 gene expression was studied at 5 h p.i. Similarly, in Fig 3E, we first monitored the expression of NF-κB and observed maximum fold increase at 5 h p.i. and selected it for further studies. Thus, ZFKM were treated with the specific inhibitors, and NF-κB gene expression was studied at 5 h p.i. 

We hope our answer addresses your concern and that the paper will be accepted.

5. Why are there two different timepoints on the same figure of Figure 4B? All the factors monitored would be best represented at the same timepoints. Further the inhibitor does not significantly increase gata-3 and klf-4 levels compared to the control (the increase was ~2-fold in Figure 4D). As such, concluding an effect of the inhibitor on these two factors is misleading when looking at the data in Figures 4C and 4D.

We are incredibly sorry for the confusion created with Fig 3 and Fig. 4. The same logic of Fig. 3 applies to Fig.4B. 
At the outset, we studied the kinetics for each of the genes to determine the time point when they are maximally expressed (Fig. 4A). It is evident from Fig.4A, that ifn-γ showed maximum expression at 12 h p.i., while tnf-α and il-4 were maximally expressed at 24 h p.i. Hence, these two-time points were selected for further studies. In the next step, ZFKM infected with M. fortuitum were treated with indicated mimics/ inhibitors, and the expression of ifn-γ monitored at 12 h p.i. and those of tnf-α and il-4 monitored at 24 h p.i. 

We appreciate your comment and have removed the data on gata3 and klf4 from the revised manuscript. Removing this data does not significantly affect the message we want to convey through this paper. We hope this addresses your concern and the manuscript is acceptable for publication.

6. Conclusions for Figures 7D and 7E are also misleading, given that the merged confocal microscopy image appears to have almost overlapping AV-stained and PI-stained cells. As mentioned earlier, this further suggests that M. fortuitim induces both necrotic and apoptotic cell death rather than selectively apoptosis. Similar conclusions were inferred in the discussion, which are not supported by the data in Figures 1 and 7.

We appreciate the concerns raised by the reviewer regarding Fig.1 and Fig.7. The results depicted in Fig.7 are an extension of what we report in Fig.1—further suggesting the pro-apoptotic role of miR-155 in M. fortuitum-pathogenesis. You will notice that in both studies, we have used AV-PI dual staining to study apoptosis/necrosis while addressing the nature of cell death induced by M. fortuitum. The same logic behind Fig.1 is applicable while answering the query of overlap noted raised by the reviewer in Fig.7. 

We sincerely hope that our explanation will be acceptable to you.

Round 2

Reviewer 3 Report

The authors have addressed some of the concerns. It would be best to include the answers to points 1 and 6 for Figures 1 and 7, respectively, in the manuscript to clarify the results. 

Author Response

We thank the reviewer for the time and effort spent reviewing our manuscript. We have included the changes suggested by you, and we hope the manuscript in its present form will be accepted for publication.